# Spontaneous Osteonecrosis of the Knee: State of the Art

**DOI:** 10.3390/jcm11236943

**Published:** 2022-11-25

**Authors:** Daniel Veloz Serrano, Samundeeswari Saseendar, Saseendar Shanmugasundaram, Rohan Bidwai, Diego Gómez, Riccardo D’Ambrosi

**Affiliations:** 1Hospital Britanico of Buenos Aires, Buenos Aires C1280 AEB, Argentina; 2Apollo Hospital Muscat, Muscat 131, Oman; 3Senior Clinical Fellow, York Teaching Hospital NHS Foundation Trust, York YO31 8HE, UK; 4IRCCS Istituto Ortopedico Galeazzi, 20161 Milan, Italy; 5Dipartimento di Scienze Biomediche per la Salute, Università degli Studi di Milano, 20133 Milan, Italy

**Keywords:** knee, spontaneous osteonecrosis of the knee, SPONK, osteoarthritis, bone marrow lesions

## Abstract

Osteonecrosis is a terrible condition that can cause advanced arthritis in a number of joints, including the knee. The three types of osteonecrosis that can affect the knee are secondary, post-arthroscopic, and spontaneous osteonecrosis of the knee (SPONK). Regardless of osteonecrosis classification, treatment for this condition seeks to prevent further development or postpone the onset of knee end-stage arthritis. Joint arthroplasty is the best course of action whenever there is significant joint surface collapse or there are signs of degenerative arthritis. The non-operative options for treatment at the moment include observation, nonsteroidal anti-inflammatory medications (NSAIDs), protective weight bearing, and analgesia if needed. Depending on the severity and type of the condition, operational procedures may include unilateral knee arthroplasty (UKA), total knee arthroplasty (TKA), or joint preservation surgery. Joint preservation techniques, such as arthroscopy, core decompression, osteochondral autograft, and bone grafting, are frequently used in precollapse and some postcollapse lesions, when the articular cartilage is typically unaffected and only the underlying subchondral bone is affected. In contrast, operations that try to save the joint following significant subchondral collapse are rarely successful and joint replacement is required to ease discomfort. This article’s goal is to summarise the most recent research on evaluations, clinical examinations, imaging and various therapeutic strategies for osteonecrosis of the knee, including lesion surveillance, medicines, joint preservation methods, and total joint arthroplasty.

## 1. Introduction

Spontaneous osteonecrosis of the knee (SPONK) is a progressive disease of the subarticular bone that can lead to subchondral collapse [1]. This pathology was first described by Ahlback et al. in 1968 as a medial femoral condyle (MFC) focal lesion, who reported its early stages as often undetected [2]. Patients with SPONK mainly complain of acute onset of pain in the knee and report no previous trauma.

Recent studies have shown an incidence of 3.4% among patients above 50 years of age and 9.4% among patients older than 65 years, with predominant affection of females compared with males by almost three to five times [3].

The clinical importance of this pathology of the knee focuses on structural changes that exert influence on the subchondral bone and articular cartilage [1,3]. Houpt et al. reported, in a review of SPONK, that the most common site of involvement is the MFC, followed by the medial tibial plateau and last, the lateral femoral condyle [4]. However, osteonecrosis (ON) of the knee usually affects a single condyle or plateau and is typically unilateral [1,3,4]. Lateral condyle SPONK is a rare condition and only a few articles have explored it. Aglietti et al. published a review examining 105 knees, reporting only one case of osteonecrosis of the lateral condyle [5]. 

The most common symptoms are localised tenderness and acute onset of pain [6,7]. Bone marrow oedema (BME) and focal subchondral lesions are cardinal signs for the diagnosis of this disease [6,7]. Yamamoto et al. suggested subchondral insufficiency fractures in the osteopenic bone as the pathogenesis of SPONK, leading to fluid accumulation in the bone marrow. However, the pathogenesis and aetiology of spontaneous knee osteonecrosis are yet to be fully understood; therefore, its management is still up for discussion [8].

Different publications have reclassified SPONK, previously considered a single entity, into three conditions: spontaneous ON of the knee; secondary ON (when the condition has a known cause); and post-arthroscopy ON [6,7].

The purpose of this literature review or descriptive paper is to evaluate the available data on the natural history and causes of SPONK.

## 2. Demographics and Classification

SPONK presents itself in patients older than 55 years of age [9]. Studies report this condition’s incidence to be 3.4% and 9.4% in persons older than 50 and 65, respectively [1]. There is no known risk factor associated with SPONK, contrary to secondary ON, to which persons younger than 50 years of age and with a history of alcohol and corticosteroid use are at risk. This disease is more prominent in women than in men [6,7]. Akamatsu et al. demonstrated a positive association between low bone mineral density and the incidence of SPONK in women over 60 years of age [10]. The male-to-female incidence is 1:5, according to Pape [11]. Al-Rowaih et al. demonstrated that the MFC was affected 94% of the time [12]. Spontaneous ON usually involves a single condyle, most often the MFC [1,3].

Koshino classified SPONK into four stages: symptoms with normal X-ray, subchondral radiolucency and osteosclerosis, subchondral collapse, and osteoarthritis (OA) [13]. This condition can also be categorised based on the size of the lesion upon X-ray examination to achieve a prognostic factor. Lesions of <3.5 cm tend to regress with no surgery, lesions of size 3.5–5 cm may or may not regress, and lesions of >5 cm lead to collapse.

The lesion can also be categorised using the Ficat classification, which describes four stages based on chondral collapse and joint space.

## 3. SPONK Aetiology

SPONK’s real aetiology is still unknown. Even though the real incidence of this disorder is not completely known, it is said to be more prevalent than secondary osteonecrosis of the knee. In contrast with SPONK, secondary ON has many known conditions and risk factors, such as sickle cell disease, myeloproliferative disorders, tobacco use and obesity, but the two most common among them are corticosteroid and alcohol abuse (>90%) [14].

Spontaneous ON is more prominent in women than in men [1,6,7]. A positive association with SPONK in women over 60 years of age and low bone mineral density supported this in a recent study by Akamatsu et al. [15].

Many publications indicate that SPONK mainly affects the medial condyle. Al-Rowaih et al., in a study of 109 patients diagnosed with spontaneous ON, identified that the medial femoral condyle was affected in 94% of the cases (102 out 109) [12].

Out of the many theories suggesting the pathogenesis for this disorder, two are the most advanced. One theory proposes a vascular affection from interference with microcirculation that produces oedema in the bone marrow, increasing pressure and further diminishing circulation until ischaemia [16]. Uchio et al. reported higher intraosseous pressure on the medial femoral condyle in SPONK patients on both femoral condyles in OA patients [17]. Reddy and Fredericks reported limited blood supply to the medial femoral condyle as opposed to the lateral femoral condyle, which has an extraosseous vascular supply [18].

Another proposed theory cites microfractures as a cause of ON. Recent evidence has demonstrated that subchondral insufficiency fractures in osteopenic bone could cause this disorder. Microfractures weaken the subchondral plate and allow joint fluid to flow from the cracked articular cartilage into the subchondral bone [19]. This may lead to the accumulation of fluid in the bone marrow, oedema with focal ischaemia, and eventual necrosis [8]. The increase in pressure may cause pain at rest or at night [19]. 

A published review of 14 patients who underwent knee surgery for SPONK by Tamamoto and Bullough evaluated the morphology of lesions, both gross and histological, and concluded that subchondral insufficiency fractures represented the primary event leading to SPONK [8]. Based on this observation, many authors consider the term SPONK as misused and suggest it be redefined as “unstable fracture resulting in bone death of the displaced fracture fragment” [14].

Post-arthroscopic osteonecrosis is considered a different category of ON, distinct from SPONK and secondary ON. Brahme et al. specifically considered arthroscopic meniscectomy as an associated cause of SPONK [20].

A systematic review of suspected aetiology found the occurrence of meniscal injuries in 50–100% of patients and suggested a strong correlation between meniscal lesions and SPONK [21].

Postmeniscectomy ON could result from subchondral bone fractures according to Higuchi et al., due to articular signal changes in bone marrow [22].

Five studies associated tears of the posterior root of the medial meniscus with SPONK [22,23,24,25,26,27]. LaPrade attributes this to the less inherent mobility of the medial meniscus because of its robust attachment to the tibia [27]. The increased femoro–tibial contact after an arthroscopic meniscectomy could cause the aforementioned subchondral insufficiency fractures due to altered lad transmission [21].

Using preoperative MRI and X-rays from 45 patients with SPONK, Yamagami et al. reported that 62.2% of SPONK patients had medial meniscus posterior root tears [28], and Robertson et al. found them in 80% of SPONK patients, assuming that ON was caused by femoral overload after meniscectomy [24].

## 4. SPONK Histology

Histology findings on SPONK were reported in a few cases in the literature, showing similar patterns: a loss of the superficial zone, vertical fissures and chondrocyte proliferation were observed in the cartilage layer. In the subchondral bone layer, articular bone plate fractures were found in addition to endochondral ossification, reactive cartilage formation and the proliferation of fibrous tissue at a depth of 3 mm from the articular surface (range: 2–5 mm). In some cases, osteonecrosis was observed in the subchondral bone of disconnected osteochondral lesions or in a thin subchondral bone attached to free cartilaginous fragments. While the deeper subchondral bone led to abnormalities, including woven bone formation and congested medullary sinuses, in some cases, no evidence of osteonecrosis was observed in this layer [29,30].

Takeda et al. reported different characteristics based on the stage of SPONK. In stage one, there is no evidence of osteonecrosis. Fibrous tissue is present around the fracture line, but the osteoid formation is not active, suggesting an early reparative reaction. In stage two, there is no evidence of antecedent osteonecrosis. However, a reparative reaction is noted, mainly consisting of osteoids and the formation of immature bone. In stage three, it is possible to see an osteonecrotic lesion confined to the area distal to the fracture line. In this case, there is inadequate repair tissue on the articular cartilage side of the fracture line, which is interpreted as a delayed union. In stage four, there are empty lacunae and necrotic debris confined to the area distal to the gap that was interpreted as showing nonunion [29].

## 5. Clinical Presentations

In primary SPONK, pain is the most common presenting feature for patients. This pain is usually localised on the affected side of the knee. The majority of patients describe the onset of the pain as spontaneous, while the pain builds up gradually in a few of them. Most patients associate it with minor knee trauma and they do recollect the exact duration of their pain [31]. Activities involving weight-bearing aggravate the pain. Progressively in the natural course of the disease, patients resort to using walking aids, and if untreated, they may end up with night pain and resting pain requiring varying doses of analgesia [32].

Knee effusion is present during the initial clinical visit and patella ballottement can be performed. The most common area with localised tenderness is over the medial femoral condyle (MFC). In the context of simultaneous involvement of both the femur and tibia, tenderness is observed in both the affected areas and gives a clue to the underlying pathology [6,7,31]. Joint range of motion (ROM) is fairly preserved apart from terminal restriction, which is painful. On valgus and varus stress, there is no evidence of instability or ligament laxity in the early stages. However, in the later stages, there is progression to most likely varus deformity and a manifestation of gait abnormalities, characterised by limp and lateral thrust. The progression to either varus or valgus deformity depends on the affected femoral condyle. MFC is the most common area and varus deformities are seen commonly in this type of patient [32].

In contrast, patients with secondary ON typically report pain that is gradual in onset and associated with risk factors related to primary aetiology. Associations with systemic lupus erythematosus and the use of corticosteroids were found to be the most common [14]. Due to the effects of multiple joints, patients usually complain of polyarthralgia, which can pose a diagnostic challenge [14].

In the context of osteonecrosis in the postoperative knee (ONKP), patients present with acute onset or exacerbation of knee pain and recent history (4–8 weeks) of arthroscopy knee surgery. As knee surgery is performed mostly for knee pain, it is usually reported by patients with concerns about the failure of knee surgery or repair damage [33]. On clinical examination, they will demonstrate a tender joint line along with effusion. However, this can be a transient normal finding in the postoperative course, and a clinician should bear this in mind for further evaluations and patient counselling [7,32,33].

## 6. Evaluation

### 6.1. Radiographs (X-rays)

These represent the first set of investigations performed for a patient suspected to have the abovementioned clinical features. Standard imaging includes anteroposterior (AP), lateral and oblique weight-bearing pain of the knee. They help in diagnosis in the later stages of the disease; however, they are negative in the early course of the disease [32]. They can also help in identifying other differentials, including occult trauma, primary osteoarthritis, and tumours.

Traditionally, in the context of SPONK, X-rays are used to decide the stage of SPONK.

Koshino et al. classified SPONK into the following four stages [13]:

Stage 1: X-ray normal.

Stage 2: Radiolucent oval shadow in the affected region.

Stage 3: Additional features of subchondral bone collapse with the formation of calcified plates and sclerotic halos.

Stage 4: Features of secondary osteoarthritis.

In the advanced stages, X-rays help in calculating the femorotibial axis, which is useful in planning correction [15].

Another system used for staging is the modified Ficat and Arlet staging. This was initially developed for femoral head osteonecrosis; however, it was subsequently applied in the context of knee osteonecrosis [15]. There is an increasing interest in calculating the femorotibial axis from long-length radiographs that help in deciding the prognosis and will be described later in the review [10,15].

To assess the established osteonecrotic lesion on X-rays, different investigators have developed methods that assist in quantifying the lesion and in deciding the treatment pathway along with effective patient counselling. These are as follows:

(A) Greatest width in AP and lateral views [10,15];

(B) The size ratio is measured by dividing the greatest AP by the width of the affected condyle [5];

(C) The surface area is calculated by multiplying the greatest AP width with the greatest lateral width;

(D) Combined necrotic angle: on AP and lateral views, the angle of the arc is tangential.

The two sides of the lesion and the fulcrum at the physical scar are measured. The sum of the two measures is equal to the combined necrotic angle. This was initially proposed for the hip by Kerboull et al. and was later modified for the knee by Mont et al. [14,34,35].

### 6.2. MRI

Because of its inherently high sensitivity, MRI is useful in the early stages of both SPONK and secondary ON. As bone oedema is presented from the early onset of the disease, these features are picked up on the MRI with high accuracy. On T1 images, features suggestive of osteonecrosis are low signal changes in the subchondral area in the affected condyle [32]. Typically, a crescentic fracture line may/can appear (in the images). These appearances were initially described in the hip and have now been identified and studied in osteonecrosis of the knee [15]. On T2-weighted images, the affected area has a margin of high signal changes [10] (Figure 1).

MRI is useful in confirming the size of the lesion initially measured on the X-ray by assessing parameters of width, depth and height of the lesion in the coronal and sagittal sections of the MRI [10].

MRI is also more reliable in measuring the extent of meniscal tears and the degree of meniscal extrusion, which was implicated in the pathophysiology of SPONK. There has been initial interest in calculating the relative percentage of meniscal extrusion in patients diagnosed with SPONK to predict the disease’s progression. After calculating the entire width of the normal meniscus, the value is compared to the degree of meniscal extrusion, thus arriving at the value of relative percentage [15]. However, in a study by Akamatsu et al., these values were not found to be significant in predicting the progression of the disease [10].

In the case of secondary ON, an MRI will show multiple lesions and the lesions can be present in either the shaft, metaphyseal region or the epiphysis as opposed to SPONK, where lesions are localised in the epiphysis in the early stage of disease [15].

While assessing the patient for osteonecrosis in the postoperative knee (ONPK), it is worth noting the finding regarding the bone marrow in the operated area of the knee. This finding, however, can be observed in various settings, such as the following: 1. frequent transient presentation after arthroscopy surgeries, including meniscectomy and ligament reconstruction; 2. SPONK presents preoperatively and is discovered in the postoperative period; and 3. true evolving ONPK [11]. In the later stages of the disease’s progression, the BME decreases and the MRI findings resemble those of SPONK with low signal areas in T1 imaging.

### 6.3. Technitium Bone Scanning

Bone scintigraphy works by detecting the uptake and has found an increased uptake in the affected condyle. They are less sensitive and specific than MRI; however, they are useful in examining patients who cannot undergo MRI. Because of the multifocal nature of secondary ON, bone scanning was thought to be sensitive; however, compared with MRI, it was found to be of limited value [14].

## 7. Associated Bony Conditions and Differential Diagnosis

In the earlier stages of the disease, MRI shows nonspecific BME in the affected area. This can also be seen in cases of occult fracture, subtle trauma (bone bruise), degenerative conditions such as osteoarthritis, and osteochondritis dissecans (adult form), as well as self-limiting hip conditions such as transient osteoporosis. This mandates a focused clinical evaluation to rule out the above similar conditions [32].

An important differential to remember when assessing BME is the case of persistent or worsening knee pain after knee arthroscopy. As suggested by Di Caprio et al. [33], to confirm the diagnosis of ONPK, BME should be absent on MRI performed preoperatively approximately 4–6 weeks after the onset of symptoms (window period for the appearance of MRI features corresponding to SPONK).

### 7.1. Low Bone Mineral Density (BMD)

The aetiology of SPONK is multifactorial, with subchondral insufficiency fractures being one of the causative factors. It is crucial to assess the presence of factors that lead to insufficiency fractures so that the development of SPONK can be predicted. Osteoporosis and osteopenia, which are highly prevalent in the geriatric population, increase the chance of insufficiency fractures [8]. Akamatsu et al. assessed the BMD of MFC for patients with an established diagnosis of SPONK and compared it with that of the control population with osteoarthritis [10]. Their study showed statistically significant low BMD in the SPONK population.

Thus, the association of low BMD supports that insufficiency fractures represent the underlying aetiological factor for SPONK.

### 7.2. Primary Osteoarthritis

As observed in studies of the natural progression of the disease, SPONK progresses to advanced osteoarthritis, which is secondary in nature. However, it is difficult to differentiate between SPONK and primary OA due to the many clinical features between them. Houpt et al. showed the existence of underlying osteoarthritis in their case series of SPONK [4]. Similar to SPONK, osteoarthritis is seen in patients above 60 years of age and most commonly affects the medial condyle. There is a significant overlap in the clinical features. However, patients with SPONK present with acute onset pain and the disease tends to progress rapidly as opposed to gradual onset and progression in osteoarthritis. Degenerative meniscus injury and cartilage damage due to preexisting OA can lead to increased load in the compartment, which predisposes the patients to subchondral fracture and progressively to SPONK. Thus, preexisting osteoarthritis of the knee can lead to the pathophysiology of SPONK [10].

## 8. Prognosis

Due to the initial normal findings on radiographs, patients with osteonecrosis of the knee tend to consult a specialist unit after a delayed period. In their study, Akamatsu et al. showed an average lag period of 49 days from the onset of symptoms before an examination by a specialist [10]. It is important to understand that the risk factors implicated in the onset of the disease, at times, do not play an equal role in the natural course of the progression of osteonecrosis of the knee.

Therefore, it is vital that the treating specialist assess these factors at the time of the initial consultation, which will help in counselling the patients and in deciding the optimal course of treatment. Akamatsu et al. analysed the impact of the femorotibial axis (FTA) on long-length radiographs and the size of the lesion on MRI as important factors in predicting the disease’s course [10]. With the clinical endpoint being conservative treatment or surgery at the end of one year since the onset of symptoms, Akamatsu et al. suggested values more likely requiring surgery with a high sensitivity and specificity of >180° on FTA and a depth of lesion >20.2 mm on sagittal MRI [10]. Interestingly, patients responding well to conservative treatment at the end of one year since the onset of symptoms tend to have a satisfactory response in the long term, which was echoed in studies by Jureus et al. and Motohashi et al. [36,37]. Inherent limitations to studies of this kind include the overlap of similar features between SPONK and those appearing because of the onset of osteoarthritis. In the future, further studies assessing patient-specific factors specifically analysing the progression of the disease will help better obtain informed consent from patients.

Recent literature underlined the importance of the area of initial osteonecrotic lesion in predicting the disease’s progression, where initial lesions >5 cm^2^ tend to fare poorly with conservative measures and progress to subsequent condylar collapse [33]. Lesions <3.5 cm^2^ tend to have a satisfactory regression of symptoms with nonsurgical methods.

It is worth noting that in the limited literature on the prognostic factors influencing the development of ONPK, the size of lesions does not help in arriving at a good prognosis as even small areas of bone marrow oedema subsequently progress to the development of osteonecrosis [11].

## 9. Treatment Strategies

Treatment strategies have evolved with the acquisition of an improved understanding of the pathophysiology; in particular, the treatment pathways depend upon the underlying aetiology and the type of osteonecrosis of the knee [15].

### 9.1. Early (Precollapse)

Conservative treatment is usually the first line of treatment in the early stages of the disease where there is no evidence of collapse on plain X-rays or if the area of osteonecrotic lesion is less than 3.5 cm^2^ [14,15]. Simple measures such as protected weight-bearing, short-term use of nonsteroidal anti-inflammatory drugs, and use of wedge insoles have shown improvement in the majority of patients if diagnosed and managed at an early stage [6,7]. There is evidence supporting the role of bisphosphonates in arresting the progression of osteonecrosis [36]. However, later studies have refuted the claim, showing no difference in the results between the use of bisphosphonates and placebo [1]. Pulsed electromagnetic field treatment is an additional treatment modality in the conservative treatment spectrum for osteonecrosis of the knee, wherein initial case series have shown encouraging results indicating the reduction of pain and size of the necrotic lesion following its administration for around six months [38].

Patients in the early stage of the disease are considered for surgical intervention when they fail non-operative treatment for 4–6 months or if the initial size of the osteonecrotic lesion is >3.5 cm^2^ [15].

Surgical procedures in the early stage of the disease depend on the degree of effect of the condyles and the underlying aetiology for osteonecrosis. Arthroscopy of the knee can be an initial minimally invasive procedure that helps the surgeon assess the lesion, classify the cartilage defect, and visualise concomitant problems (meniscal extrusion, tears, etc.). Meniscal tears can be addressed in the same sitting. Additional procedures, such as debridement, microfracture and core decompression, can be performed along with arthroscopy. This step also helps the surgeon in determining prognosis and in planning future treatment if there is evidence of full-thickness cartilage defects in multiple areas of the knee or evidence of multiple degenerative changes [1]. In a study on 16 patients, Forst et al. showed pain relief and improved clinical outcomes after core decompression in patients with refractory SPONK [39]. Core decompression involves percutaneous drilling of the affected condyle and helps reduce the increased marrow pressure, thereby improving the vascular channel and arresting the progression of the disease. Arthroscopic microfracture involves intra-articular drilling through the cartilage defect into the bone marrow, thereby introducing marrow fluid and blood to encourage healing, which can potentially improve pain outcomes. Arthroscopic microfracture and debridement have also shown improved outcomes in early SPONK at a mean follow-up of 27 months postoperatively [40].

An inherent drawback of these studies is that although they have encouraging short-term outcomes, the long-term benefits of these procedures remain doubtful. Further studies focusing on long-term radiological and clinical outcomes will help improve decision-making.

### 9.2. Late Stage

In the progression of disease pathology, Ficat stage three is an important stage, as it transitions a surgeon from a joint-preserving option to joint replacement options. Unicompartmental knee arthroplasty (UKA) has shown significant pain relief and improved outcomes in appropriately indicated primary osteoarthritis patients, and these results have persisted in the long term. With a current vogue on having increased usage of UKA for adult reconstruction surgery and improved understanding, it is a valuable tool in the context of SPONK having the affection of a single condyle [6,7]. Heyse et al. have shown improved pain and clinical scores that underline their significance [41]. However, when the disease process involves the other compartment, total knee arthroplasty (TKA) remains the most reasonable and predictable option, offering satisfactory pain relief and improved function. Studies have shown comparative long-term outcomes when TKA is offered for the end stage of SPONK compared with those in whom it was offered for primary osteoarthritis [42].

A viable alternative to UKA in a young patient with a high activity level is high tibial osteotomy, which offloads the affected condyle by altering the weight-bearing axis. Koshino et al. published the results of improved clinical outcomes in a series of 37 knees at a mean follow-up of 62 months [37]. The last decade, however, has witnessed an increased interest in UKA for unicompartmental affection. However, with the advent of modern implants and a better understanding of biomechanics, high tibial osteotomy is an effective joint-preserving option in an appropriately indicated patient.

The osteochondral option can be the bridging option between joint preservation and joint replacement options, wherein the procedure involves grafting the defect by harvesting an osteochondral graft from a non-weight-bearing portion in an attempt to restore the articular cartilage and thereby preserve the native joint, as described by Hangody et al. [43]. There are very few studies focusing on the results of osteochondral grafting for SPONK. Tanaka et al., in their case series of six patients, demonstrated satisfactory pain relief at a mean follow-up of 28 months.

The management principles for secondary ON differ, as there is a low threshold for the consideration of surgical procedures in the early precollapse stage if the underlying aetiology cannot be reversed. The early precollapse stage fares poorly with conservative measures and is a candidate for the consideration of surgical options [15]. In the late stages of the natural course of the disease, because of the usual multicondylar involvement in secondary ON, TKA is preferred over UKA.

In the context of ONPK, the logical first step in treatment would be conservative management and avoiding another surgery. Conservative management would include protected weight-bearing, analgesia and follow-up MRI to assess the development or progression of lesions. Pape et al. showed poor outcomes following conservative management in their series of 47 patients with ONPK [11]. As discussed in prognosis, the size of the lesion does not determine progression, and early diagnosis of the disease would increase the chance of success of conservative management. However, in the event of failure of conservative management or the passage of six months since the knee surgery, surgical management can be contemplated based on the guidelines for SPONK.

## 10. Conclusions

Osteonecrosis of the knee can come in three different forms: spontaneous, secondary, and post-arthroscopic. Similar to osteonecrosis of the hip, secondary osteonecrosis is linked to corticosteroid and alcohol usage. It may also be brought on by vascular obstruction and marrow crowding. Fractures resulting in subchondral insufficiency may be the cause of spontaneous and posttraumatic osteonecrosis. The majority of these illnesses are diagnosed and staged using plain radiographs and MRI, with MRI being the most sensitive and precise diagnostic tool. While secondary osteonecrosis is best addressed surgically, spontaneous and post-arthroscopic osteonecrosis of the knee can initially be managed non-operatively. Joint preservation surgeries are used as the initial surgical therapy for patients who are in the precollapse stages, while arthroplasty is used as the final surgical therapy for symptomatic patients who have severe osteoarthritis and collapse.

## Figures and Tables

**Figure 1 jcm-11-06943-f001:**
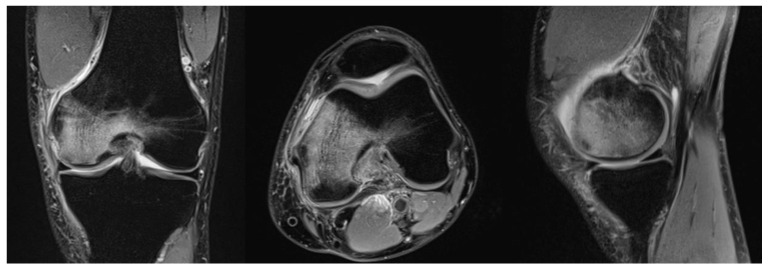
MRI of a right knee in a 51-year-old man showing osteonecrosis of the femoral condyle.

## Data Availability

Not applicable.

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
