# Peer review of "Spontaneous Osteonecrosis of the Knee: State of the Art"

_jcm, 2022, doi:10.3390/jcm11236943_

Round 1

Reviewer 1 Report

SPONK is a progressive disease involving the osteochondral complex, and its aetiology is still unknown. In this review paper, the authors summarized the demographics and classification, aetiology, histology, clinical presentations, investigations, differential diagnosis, prognosis and treatment strategies of SPONK. This review is informative, but there are some major issues concerning the manuscript. In the aspect of language, it is a little tedious, repetitive and unprofessional in some parts of the manuscript, which could be improved. The Abstract could be more informative and logical, highlighting the key points of the specific review paper. Concerning the structure of the main manuscript, the paragraphing is too causal. More importantly, no innovative information and attracting viewpoints are provided, making the paper more like a chapter of a book. In addition, there is no conclusion section.

Author Response

We have reviewed all the manuscript according to your suggestions

Reviewer 2 Report

The authors have organized a relatively good flow and composition of knee SPONK, so that readers can acquire appropriate clinical knowledge about it.

It is thought that 'etiology' is a more commonly used word than 'aetiology'.

line 46: delete 'physical'.

line 46: localised -> localized 

line 60: 3,4% -> 3.4%/ 9,4%->9.4%

line 69: X-ray -> x-ray

line 124: 62,2% -> 62.2%

line 157: night pain/rest pain -> night pain and resting pain 

line 168: If you are going to use MFC as an abbreviation, write the same when mentioned later.

line 182: It is considered that "evaluation" is more suitable than "Investigations".

line 186: delete 'film'

line 189: tumour -> tumor

Line 190: SONK? Is it a typo to suddenly use a term? Or did you mean to write SONK? There is no explanation for the abbreviation.

Line 202: Delete the expressions in parentheses.

Line 208: If you use the abbreviation (AP), write it the same before and after.

Line 220,232, 234: recheck 'SONK'

It is thought that it will be more beneficial if the contents of CT are added in Evaluation.

Line 264: Recheck and express the BME in the same place where the front and back overlap.

Line 268: Please describe your BMD as full name

Line 315, 325: It is considered that the names of the authors of the cited papers are unnecessary.

Line 331: It is considered that the abbreviation is not necessary for NSAIDs that have been described only once.

Line 375: TKA? please describe full name once. 

Line 386-387: delete 'hyphen'

Please add the conclusion of your paper following the last paragraph. 

Author Response

The authors have organized a relatively good flow and composition of knee SPONK, so that readers can acquire appropriate clinical knowledge about it.

It is thought that 'etiology' is a more commonly used word than 'aetiology'.

Changed through all the manuscript

line 46: delete 'physical'.

Removed as suggested

line 46: localised -> localized

changed as suggested

line 60: 3,4% -> 3.4%/ 9,4%->9.4%

changed as suggested

line 69: X-ray -> x-ray

changed as suggested

line 124: 62,2% -> 62.2%

changed as suggested

line 157: night pain/rest pain -> night pain and resting pain

changed as suggested

line 168: If you are going to use MFC as an abbreviation, write the same when mentioned later

corrected through all the manuscript

line 182: It is considered that "evaluation" is more suitable than "Investigations".

Changed as suggested

line 186: delete 'film'

removed as suggested

line 189: tumour -> tumor

changed as suggested

Line 190: SONK? Is it a typo to suddenly use a term? Or did you mean to write SONK? There is no explanation for the abbreviation.

Changed through all the manuscript

Line 202: Delete the expressions in parentheses.

Removed as suggested

Line 208: If you use the abbreviation (AP), write it the same before and after.

Corrected as suggested

Line 220,232, 234: recheck 'SONK'

Changed as suggested

It is thought that it will be more beneficial if the contents of CT are added in Evaluation.

 We included mri figure

Line 264: Recheck and express the BME in the same place where the front and back overlap.

Changed through all the manuscript

Line 268: Please describe your BMD as full name

 corrected

Line 315, 325: It is considered that the names of the authors of the cited papers are unnecessary.

 Removed and rephrased

Line 331: It is considered that the abbreviation is not necessary for NSAIDs that have been described only once.

Removed as suggested

Line 375: TKA? please describe full name once.

Added as requested

Line 386-387: delete 'hyphen'

 Removed as suggested

Please add the conclusion of your paper following the last paragraph.

Added as suggested

Round 2

Reviewer 1 Report

Most important concerns are not solved.

Although this is a narrative review, it is pivotal to provide more new insights or progress in this area. Otherwise, we could refer to a text book in which SPONK is described. From this point, it is not a qualified academic review.